# Dry Reforming of Methane over Li-Doped Ni/TiO₂ Catalysts: Effect of Support Basicity

**Vicente Pérez-Madrigal** [1], **Edna Ríos-Valdovinos** [1], **Elizabeth Rojas-García** [2], **Miguel A. Valenzuela** [3] **and Francisco Pola-Albores** [1,*]

[1] Laboratorio de Materiales y Procesos Sustentables, IIIER—Universidad de Ciencias y Artes de Chiapas, Libramiento Nte. Pte. 1150. Col. Lajas Maciel, 29039 Tuxtla Gutiérrez, Chiapas, Mexico
[2] Área de Ingeniería Química, Universidad Autónoma Metropolitana-Iztapalapa, Av. San Rafael Atlixco 186, Col. Vicentina, 09340 Ciudad de México, Mexico
[3] Laboratorio de Catálisis y Materiales, ESIQIE—Instituto Politécnico Nacional, Zacatenco, 07738 Ciudad de México, Mexico
[*] Correspondence: francisco.pola@unicach.mx; Tel.: +52-61-70440 (ext. 4299)

**Abstract:** In this research, we investigate the impact of Li doping on a TiO₂ support, synthesized through the sol-gel method, with a focus on varying the aging time. Our objective is to elucidate how aging duration and doping influence the surface basicity, thereby mitigating carbon formation and amplifying the catalytic efficacy of Ni-loaded catalysts (15 wt.%). Essential characterization techniques encompass X-ray diffraction, H₂-TPR, FE-SEM, N₂-physisorption, DLS, FTIR, and Raman spectroscopies. Our findings reveal that extended aging periods promote the development of a basic character, attributable to oxygen defects within TiO₂. This inherent trait bears significant implications for catalyst performance, stability, and carbon formation during the reaction. Remarkably, the catalyst with the highest catalytic activity and stability boasts an 85% relative basicity, a property also induced by incorporating lithium into the TiO₂ support.

**Keywords:** dry reforming of methane; supported Ni catalysts; Li-doped TiO₂; surface basicity; aging time effect





## 1. Introduction

The world's energy consumption is mainly based on fossil fuels for industrial, domestic, and transportation applications, so greenhouse gas production is very high. Carbon dioxide, among others, is one of the most responsible for causing climate change [1,2]. This situation drives an exhaustive exploration of energy use and utilization alternatives, even mitigating CO₂ technologies. In such a way, capturing CO₂ emissions and converting them is strongly necessary to stabilize global warming. However, humanity's efforts are going in the opposite direction: the world's temperature is going towards a catastrophic increase that will have devastating consequences for life on the planet [3]. Remarkably, the dry reforming of methane (DRM) has had a growing interest in the last decade, because it uses two leading greenhouse gases (CO₂ and CH₄) as reactants to obtain syngas (CO and H₂), according to Equation (1).

$$CH_4 + CO_2 \rightarrow 2CO + 2H_2 \quad \Delta H_R^\circ = 247.3 \text{ kJ/mol} \tag{1}$$

The DRM can use industrial CO₂ waste emissions, biogas, and natural gas to transform into syngas, an essential raw material for manufacturing useful value-added products (e.g., hydrogen, methanol, and liquid fuels) [4,5]. DRM has been studied with various catalysts, including noble and transition metals deposited on supports, such as SiO₂, MgO, CeO₂, La₂O₃, Al₂O₃, TiO₂, or mixed oxides, and adding metal promoters. Nevertheless, Ni-based catalysts represent a promising alternative for methane activation and are low cost, as long as their current problems, associated with coke formation and sintering, are

resolved [6]. Some reactions leading to carbon formation are present in DRM, such as methane cracking (Equation (2)) and the Boudouard reaction (Equation (3)), among others (Equations (4) and (5)), which could drive catalyst deactivation:

$$CH_4 \rightarrow C + 2H_2 \tag{2}$$

$$2CO \rightarrow C + CO_2 \tag{3}$$

$$2H_2 + CO_2 \rightarrow C + 2H_2O \tag{4}$$

$$H_2 + CO \rightarrow C + H_2O \tag{5}$$

DRM is endothermic ($\Delta H_R^\circ = 247.02$ kJ/mol). Thermodynamically, the reaction occurs at temperatures above 643 °C (at 25 °C, $\Delta G^\circ = 170.53$ kJ/mol); the rest of the parallel reactions are exothermic, except for methane catalytic cracking [7].

The stability of Ni-based catalysts has been improved using supports and promoters with a high Lewis base character, leading to carbon inhibition and, consequently, diminishing deactivation [8,9]. This fact drives into the concept that basicity variation could be a way to improve activity and reduce deactivation. Several authors have shown an intrinsic correlation between the basicity of the metal oxide support and catalytic performance [10–17]. Furthermore, it has been widely reported that the chemical behavior of $TiO_2$ is determined by the Lewis base character, which is due to the presence of $O^{-2}$ sites on its surface [18]. DFT studies on rutile (110) planes have demonstrated high levels of interaction with $CO_2$ molecules in vacancy sites [19,20]. Oxygen vacancies in $TiO_2$ are electron pair donors (donor states under a conduction band) [20–22]. Theoretical studies show that anatase (101) and (001) present a lower defect concentration compared with rutile (110) [23]. Several works have published the use of $TiO_2$ as a catalyst support in reactions, taking advantage of its basicity and thermal and chemical stability. Aging time in $TiO_2$ materials has been reported to increase the conversion efficiency in solar cells [24,25], but not in DRM.

For example, synthesized $TiO_2$, with nanosheet morphology, supporting a Ni-based catalyst showed a high conversion of $CH_4$ and $CO_2$ and a good carbon dissociation over the catalyst surface [26]. Also, modified $Co/TiO_2$ with Pt, Ru, and Ni achieved slow deactivation [27]. Some other works have reported the basicity of $Ni/La_2O_3$ catalysts with a nanorod morphology, showing higher efficiencies in increasing basic sites [28]. Furthermore, weak basic sites have been reported to be helpful for DRM through using $CaO_2$–$ZrO_2$ catalysts [29]. Regarding the alkaline-$TiO_2$ system, it has been demonstrated that alkali atoms adsorbed on the $TiO_2$ surface, such as Li, Na, or K, behave as Lewis base sites because they donate electron pairs [20]. Lithium-doped $TiO_2$ materials have also found applications as energy storage materials, in photocatalysis, and as catalysts in the transesterification of bisphenol-A [30,31].

As a probe molecule, surface basicity can be monitored with MBOH (2-Methyl-3-butyl-2-ol). MBOH dehydrates into isopropyl acetylene, forming acetone and acetylene on basic sites [32,33]. This technique is feasible for evaluating the acid-base properties in solids [34].

This work correlates the effect of the basicity of the $TiO_2$ (support) due to aging time and the corresponding catalytic activity and stability in the DRM, keeping constant the composition of the active phase (Ni particles) and promoter (Li).

## 2. Results and Discussion

### 2.1. Phase Analysis of Calcined Catalysts

The X-ray diffraction patterns corresponding to the Li-doped $Ni/TiO_2$ calcined catalysts are compared in Figure 1. The summarized structural and refinement details are presented in Table 1. The presence of rutile (PDF 021-1276, s.g. 136), anatase (PDF 021-1272, s.g. 141), nickel (II) titanate (PDF 033-0960, s.g. 148), and nickel oxide (bunsenite) (PDF 047-1049, s.g. 225) were observed. The quantitative phase analysis is shown in Table 2. According to these results, anatase was the main component in all catalysts, ranging from

63.5 to 72.7 wt.%. Note that a higher amount of anatase $TiO_2$ was found with the Ni/T4, i.e., after five days of aging. Furthermore, the binary metal oxide $NiTiO_3$ phase was observable in all samples, with the most significant amount being observed in the Ni/T4 catalyst. Lithium phases were not detected.

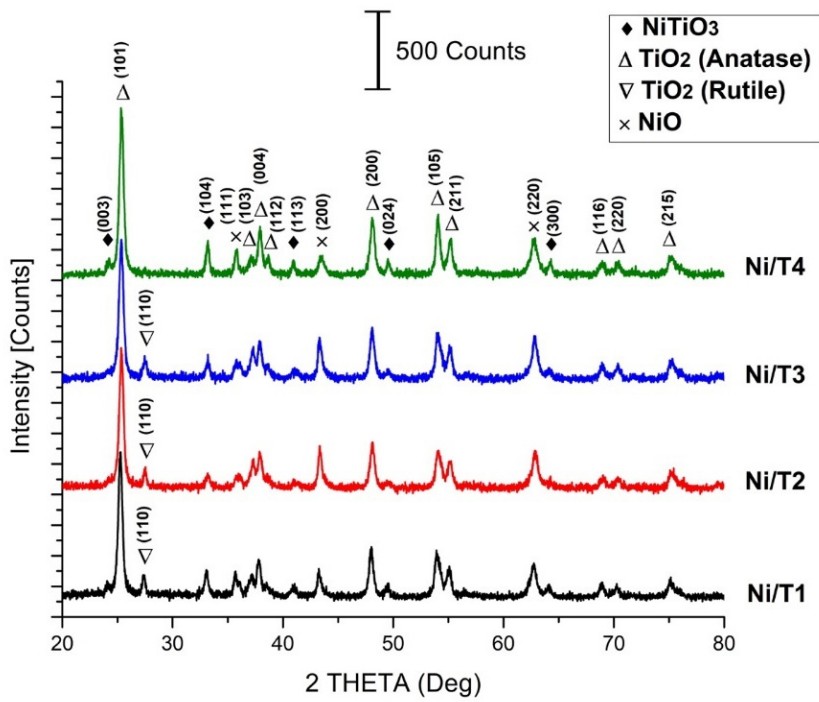

**Figure 1.** X-ray diffraction patterns of Li-doped $Ni/TiO_2$ calcined catalysts.

**Table 1.** Structural and Rietveld refinement details for all involved phases in fresh and spent catalysts.

| Sample | Input Structural CIF File | Space Group | Lattice Parameters [nm] | | Atomic Positions | | | General FullProf Details |
|---|---|---|---|---|---|---|---|---|
| | | | *a* | *c* | *x/a* | *x/b* | *x/c* | |
| Rutile | 647553 | 136 | 0.458 | 0.295 | Ti = 0.00 | Ti = 0.00 | Ti = 0.00 | |
| | | | | | O = 0.305 | O = 0.305 | O = 0.00 | |
| Anatase | 9852 | 141 | 0.37842 | 0.95146 | Ti = 0.00 | Ti = 0.00 | Ti = 0.00 | Peak shape function: Npr = 7, TCH pseudo-Voigt |
| | | | | | O = 0.00 | O = 0.00 | O = 0.2081 | Resolution function type: |
| $NiTiO_3$ | 33854 | 148 | 0.50274 | 1.3783 | Ni = 0.00 | Ni = 0.00 | Ni = 0.3507 | RES = 4, Instrumental resolution file |
| | | | | | O = 0.316 | O = 0.0149 | O = 0.2471 | Occupancies: |
| | | | | | Ti = 0.00 | Ti = 0.00 | Ti = 0.1445 | Initially, set to 1. Refined. |
| NiO | 9866 | 225 | 0.478 | 0.478 | Ni = 0.00 | Ni = 0.00 | Ni = 0.00 | Isotropic displacement (temperature) |
| | | | | | O = 0.500 | O = 0.500 | O = 0.500 | parameter: |
| $LiTi_2O_4$ | 15789 | 227 | 0.83910 | 0.83910 | Li = 0.0000 | Li = 0.000 | Li = 0.000 | Not considered |
| | | | | | O = 0.3900 | O = 0.3900 | O = 0.3900 | |
| | | | | | Ti = 0.6250 | Ti = 0.6250 | Ti = 0.6250 | |
| Ni | 52331 | 225 | 0.35243 | 0.35243 | Ni = 0.00 | Ni = 0.00 | Ni = 0.00 | |
| C | 31170 | 186 | 0.2470 | 0.6790 | $C_1$ = 0.000 | $C_1$ = 0.000 | $C_1$ = 0.000 | |
| | | | | | $C_2$ = 0.3333 | $C_1$ = 0.6667 | $C_1$ = 0.005 | |

**Table 2.** Quantitative phase analysis obtained by XRD of Li-doped Ni/TiO$_2$ calcined catalysts (in wt.%), the NiO crystallite size, and the anatase (A) to rutile (R) ratio.

| Sample | Anatase | Rutile | NiO | NiTiO$_3$ | NiO Crystallite Size [nm] | A/R Ratio |
|--------|---------|--------|------|-----------|----------------------------|-----------|
| Ni/T1  | 66.4    | 8.6    | 15.7 | 9.3       | 23.8                       | 7.72      |
| Ni/T2  | 68.7    | 7.8    | 8.6  | 14.9      | 19.8                       | 8.81      |
| Ni/T3  | 63.5    | 10.3   | 10.9 | 15.3      | 22.7                       | 6.17      |
| Ni/T4  | 72.7    | 1.6    | 17.4 | 8.3       | 23.3                       | 45.44     |

### 2.2. Morphology Analysis Using Scanning Electron Microscopy

Figure 2a–d shows the scanning electron microscopy (SEM) results of the Li-doped Ni/TiO$_2$ catalysts.

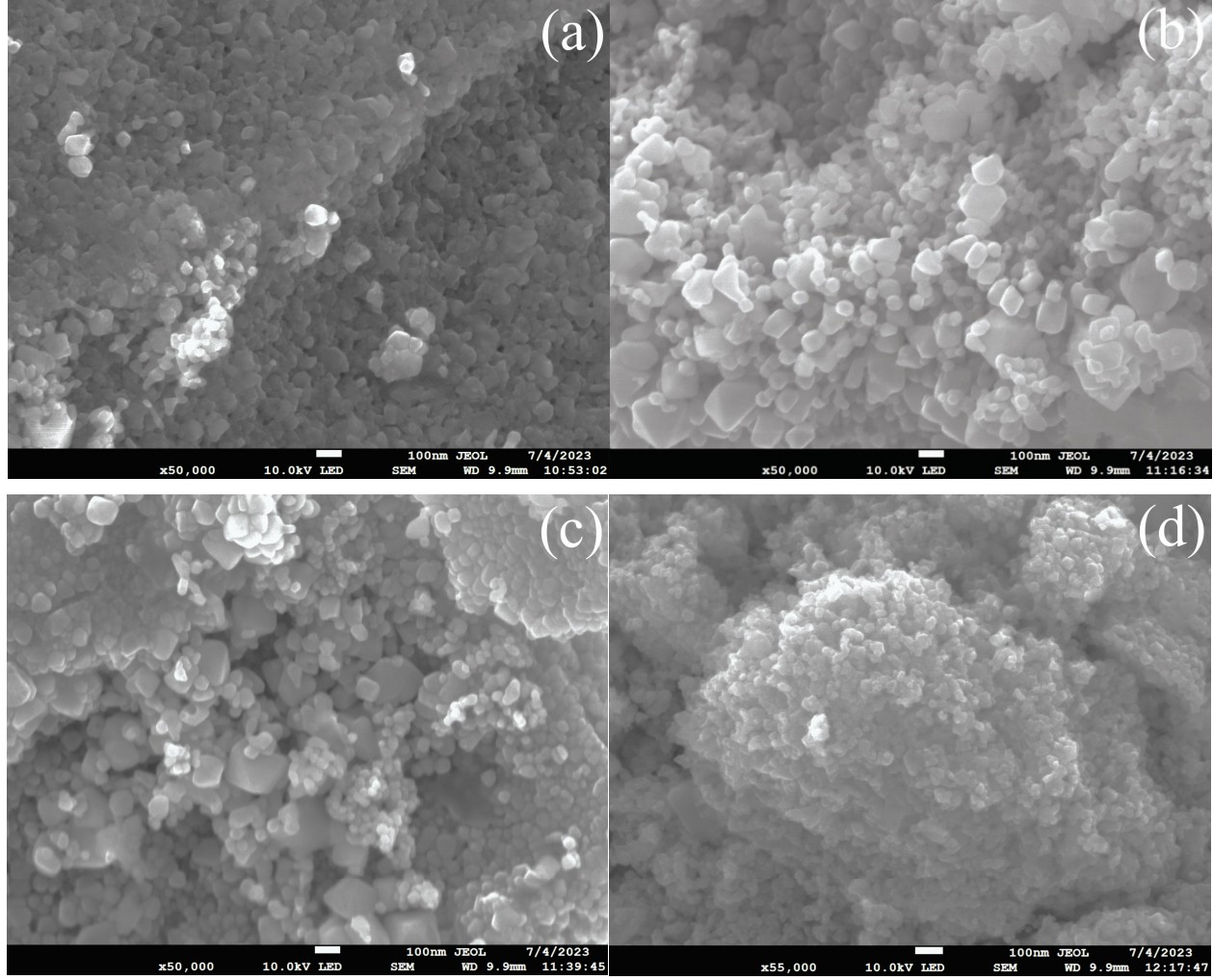

**Figure 2.** FESEM micrographs of the Ni-Li/TiO$_2$ catalysts: (**a**) Ni/T1, (**b**) Ni/T2, (**c**) Ni/T3, and (**d**) Ni/T4 samples.

Images show round-shaped particles in all cases, all attributable to the NiO phase (no EDS analysis was performed). For Ni/T1 and Ni/T2 samples shown in Figure 2a,b, the particles range from 30 to 100 nm, and in the first case, they are visibly compacted. In the Ni/T3 sample shown in Figure 2c, bigger particles appear, ranging from 100 to 200 nm, and are differentiated from those around 50 nm or less. Finally, the Ni/T4 sample shown in Figure 2d shows a narrow particle size distribution, between 30 and 50 nm, suggesting a better active phase dispersion in this catalyst.

### 2.3. Temperature-Programmed Reduction

TPR results are shown in Figure 3 and Table 3. The peaks observed in the two regions (I and II), of 330–366 °C and 414–440 °C, are attributed to reductions of different NiO species, depending on their strong or weak interactions with the TiO$_2$ support. The H$_2$ uptake in region I is attributed to NiO single-phase reduction (bulk) [35], while the peaks in region II (414–440 °C) are attributed to NiO interacting with the TiO$_2$ surface as NiO-TiO$_2$ [35,36]. It has been discovered that the interaction of NiO and TiO$_2$ as anatase is considerably weaker than that with TiO$_2$ as rutile [37,38]; according to this degree of interaction, H$_2$ consumption fluctuates in region II. The last region (III) represents the less reducible Ni$^{2+}$ in NiTiO$_3$ particles, and its presence is demonstrated in X-ray measurements. According to the material balance in Table 3, calculated over the XRD results base, the H$_2$/Ni$^{2+}$ ratio is under the stoichiometric ratio in all cases. This could only suggest that the overall NiO and Ni$^{2+}$ in NiTiO$_3$ phases in the catalysts were not wholly reduced by particle size or mass transport effects. NiO reduction by TPR analysis ranges from 60.0 to 79.9%, while the reducibility of NiTiO$_3$ to metallic Ni oscillates from 35.4 to 85.6%.

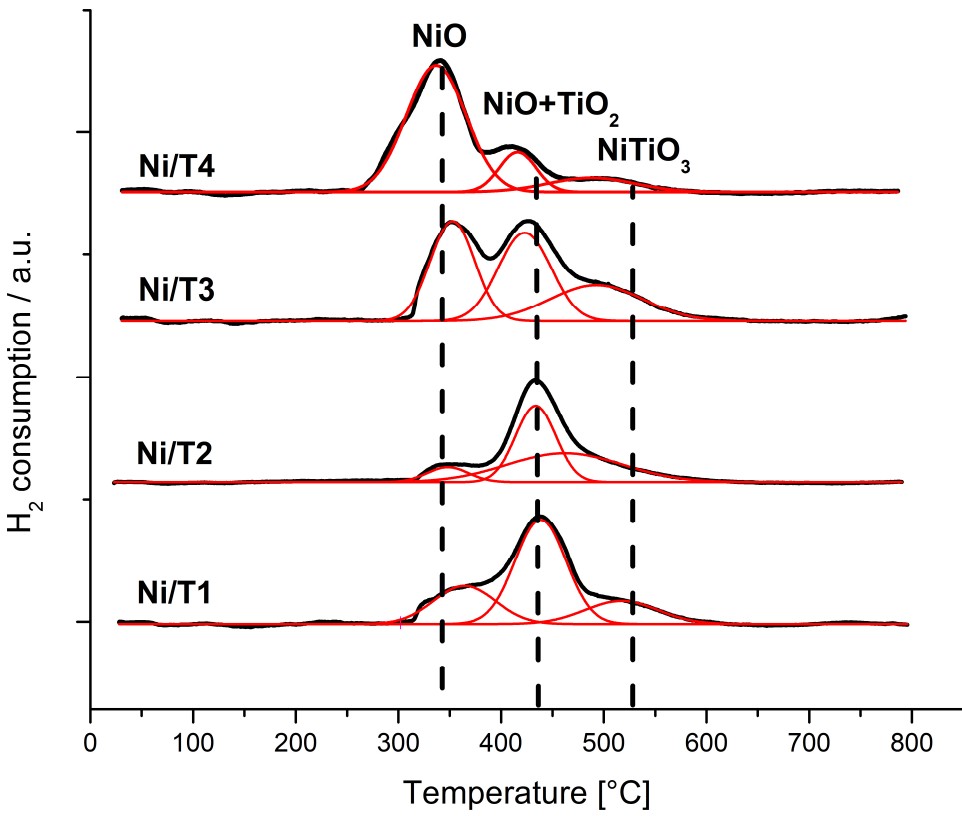

**Figure 3.** H$_2$-TPR profiles of Li-NiO/TiO$_2$ catalysts. (Red lines: profile deconvolution according to regions I, II and III).

**Table 3.** H$_2$-TPR results for fresh catalysts.

| Sample | Total H$_2$ Consumption (μmol/g$_{cat}$) | Total Ni$^{2+}$ (μmol/g$_{cat}$) | H$_2$/Ni$^{2+}$ | NiO, NiTiO$_3$ Molar Ratio | Total NiO Reduced (%) | Total NiTiO$_3$ Reduced (%) | Peak Center (°C), by Regions ($\beta$ = 10 °C min$^{-1}$, X$_{H2}$ = 5%) | | |
|---|---|---|---|---|---|---|---|---|---|
| | | | | | | | I | II | III |
| Ni/T1 | 1593 | 2704 | 0.59 | 3.5 | 62.2 | 47.4 | 366 | 440 | 519 |
| Ni/T2 | 1745 | 2115 | 0.82 | 1.2 | 79.9 | 85.6 | 351 | 435 | 464 |
| Ni/T3 | 1518 | 2449 | 0.62 | 1.5 | 75.9 | 41.4 | 353 | 424 | 495 |
| Ni/T4 | 1588 | 2867 | 0.55 | 4.3 | 60.0 | 35.4 | 334 | 414 | 485 |

*2.4. DRIFTS Analysis*

Figure 4a–d shows the IR spectra of saturated samples with MBOH. Figure 4a displays an asymmetric band at 1462 cm$^{-1}$. Moreover, a symmetric band is observed at approximately 1371 cm$^{-1}$, which can be attributed to a methyl group linked to a carbon atom. The group of t-butyl is confirmed at 1215 cm$^{-1}$, while the isopropyl group could appear at 1178 cm$^{-1}$. The vibration at 3500–3400 cm$^{-1}$ is attributed to the C=O. Figure 4b shows the band at 3645 cm$^{-1}$, which shows the OH- groups produced in the catalyst [32]. Figure 4c indicates an extension of the bond C-H could be found at 2995 cm$^{-1}$, and the asymmetric and symmetric peaks of CH$_3$ and CH$_2$ are located at 2931 and 2857 cm$^{-1}$, respectively. These bands demonstrate the MBOH linked to the catalyst's surface. Additionally, bands at 1748, 1672, and 1612 cm$^{-1}$, corresponding to C=O, CO (amide I), and NH$_2$ (amide II) bonds, respectively, can be seen in Figure 4d [39]. Due to their basic character, these new bands ratify the acetylene group formation after linking MBOH on the catalyst's surface. Therefore, the areas under these bands' curves are directly related to the surface basicity of the catalysts; the quantitative determination for acetylene and OH$^-$ groups is reported in Table 4. Analysis with MBOH of 100 wt.% anatase and 100 wt.% rutile used as reference materials was used to compare the contribution of these phases to basicity. As expected, rutile scarcely contributes to the basic character of the catalysts; only the anatase has a notable effect.

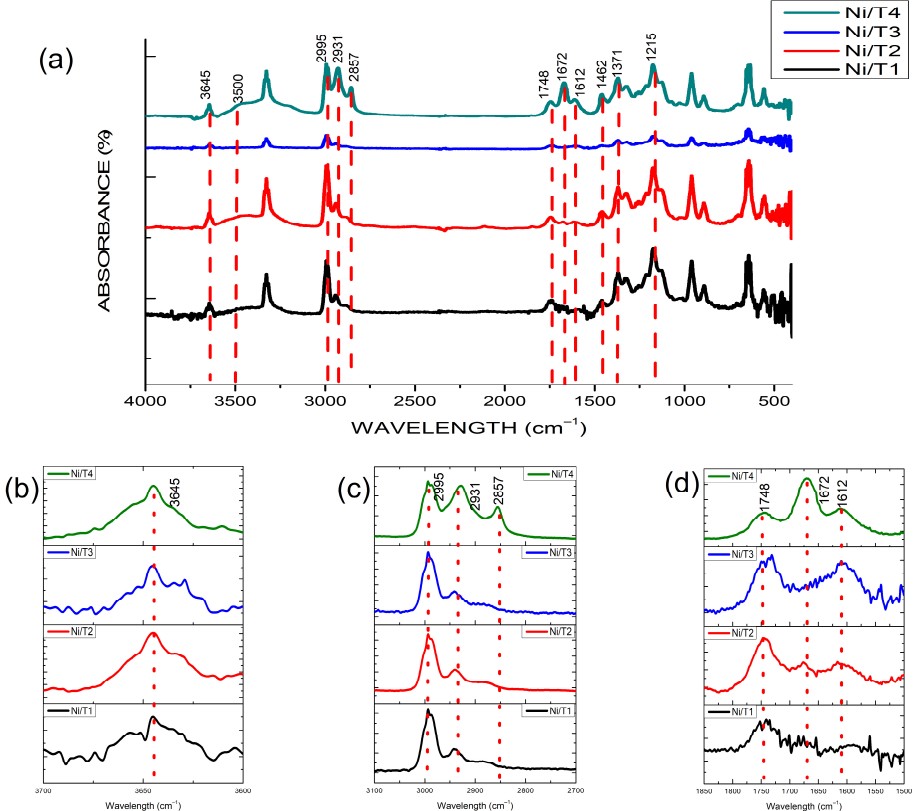

**Figure 4.** DRIFTS measurements of fresh catalysts at 70 °C with MBOH. Regions: (**a**) 4000 to 500 cm$^{-1}$, (**b**) 3700–3600 cm$^{-1}$, (**c**) 3500–2500 cm$^{-1}$, and (**d**) 2000–1500 cm$^{-1}$.

The increasing basicity order was as follows: Ni/T4 > Ni/T2 > Ni/T1 > Ni/T3. It is worth noting that, as the anatase phase increases, the higher the observed surface basicity is. As we will see later, surface basicity also plays an essential role in the DRM catalytic activity, mainly reducing carbon deposition.

Figure 5 shows CO–DRIFTS chemisorption measurements at 200 °C on the reduced catalysts. They have not observed the CO adsorption bands at 2035 cm$^{-1}$ for linearly bonded CO and at 1925 and 1775 cm$^{-1}$ for multifold adsorbed CO [40]. Despite 12 wt.%

Ni loading (obtained from Table 2) in the CO atmosphere, no detectable IR signal was collected, suggesting a strong metal–support interaction (SMSI) effect. The IR band at 1603 cm$^{-1}$ is associated with a formate species (OCO) [41].

**Table 4.** Areas under the curve of acetylene ($A_{acet}$), OH groups ($A_{OH}$), and relative basicity (%), determined from the results obtained by DRIFTS (Figures 3 and 4). * Reference materials.

| Sample | $A_{acet}$ (1612, 1672 y 1748 cm$^{-1}$) | $A_{OH}$ (3645 cm$^{-1}$) | Relative Basicity (%) |
|---|---|---|---|
| Rutile * | 1.142 | 1.084 | 5.5 |
| Ni/T1 | 4.223 | 1.042 | 20.3 |
| Ni/T2 | 4.92 | 1.3 | 23.6 |
| Ni/T3 | 2.225 | 0.52 | 10.7 |
| Ni/T4 | 17.628 | 1.763 | 84.7 |
| Anatase * | 20.808 | 3.401 | 100.0 |

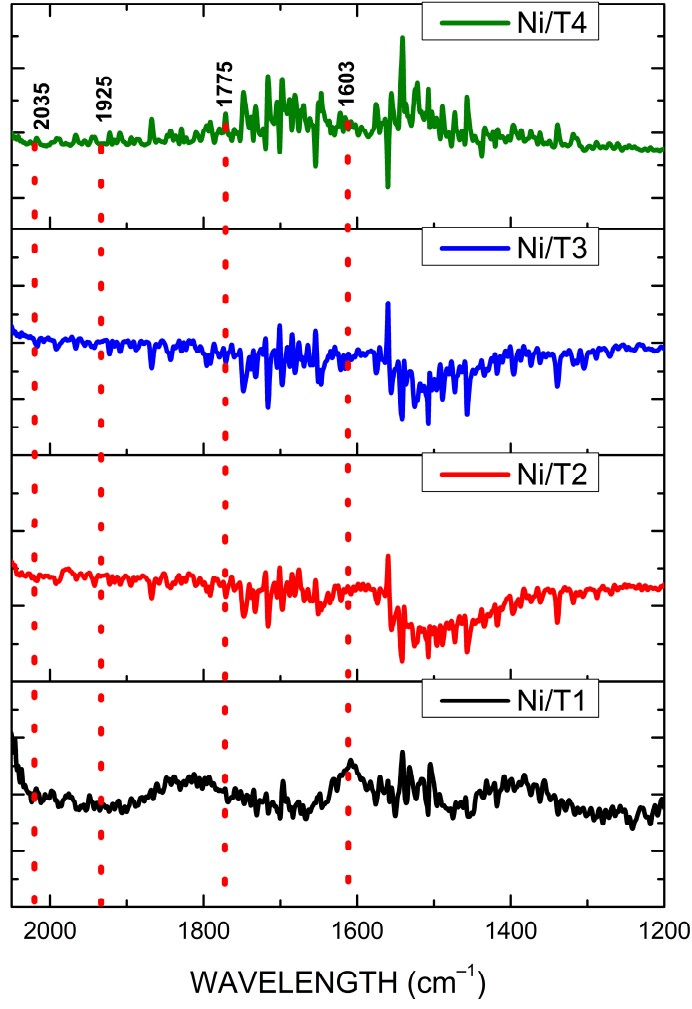

**Figure 5.** DRIFTs analysis of CO over the reduced catalysts' surfaces. Measurements at 200 °C with 10% CO (balanced with Ar).

*2.5. Particle Size and Textural Properties*

The particle size and textural properties are shown in Table 5. It is observed that dynamic particle size follows the order: Ni/T4 < Ni/T2 < Ni/T1 < Ni/T3. It is also observed that the BET area only varies at a shorter aging time and is subsequently preserved. In the case of pore diameter, it is shown that the size remains constant in the samples Ni/T1 to Ni/T3 and only in Ni/T4 decreases, which is the catalyst with the highest efficiency in DRM.

While regarding the pore volume, it is observed that the samples Ni/T1 and Ni/T4 have the smallest volumes. Figure 6 shows the adsorption–desorption isotherms of different Ni/TiO$_2$ catalysts. It is observed that all catalysts show isotherms of type IV, according to the classification of the IUPAC, which means that the materials are mesoporous, due to the presence of TiO$_2$.

**Table 5.** Particle size and textural properties of fresh catalysts.

| Sample | Particle Size [nm] | S $_{BET}$ [m$^2$ g$^{-1}$] | Pore Diameter [nm] | V$_P$ [cm$^3$ g$^{-1}$] |
|---|---|---|---|---|
| Ni/T1 | 730.9 | 11.0 | 14.31 | 0.08 |
| Ni/T2 | 437.9 | 20.5 | 14.30 | 0.11 |
| Ni/T3 | 748.2 | 20.1 | 14.58 | 0.11 |
| Ni/T4 | 384.3 | 19.7 | 8.51 | 0.07 |

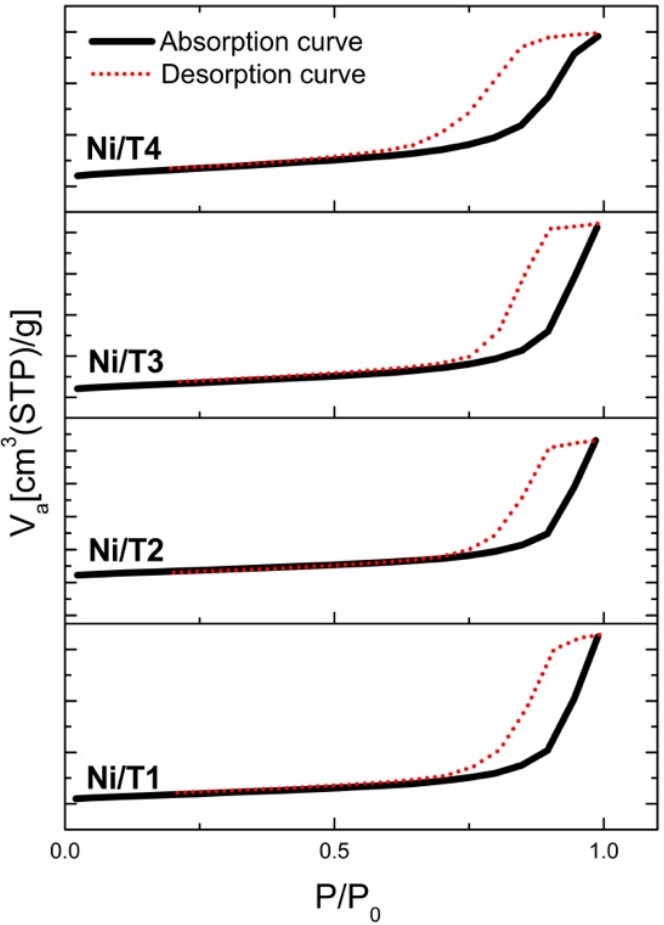

**Figure 6.** Adsorption and desorption isotherms of Li-doped Ni/TiO$_2$ catalysts.

### 2.6. Raman Spectroscopy of Calcined Catalysts

Figure 7a shows the Raman spectra of samples (synthesized at different aging times). In all samples there are observed bands at 144 (E$_g$), 398 (B1$_g$), 514 (B1$_g$ + A1$_g$), and 636 cm$^{-1}$ (E$_g$); all of these are characteristic of the TiO$_2$ anatase phase [42]. There are also bands at 242 (corresponding to second order effect, SOE), 441 (E$_g$), and 608 (A1$_g$) cm$^{-1}$, indicative of the TiO$_2$ rutile phase [43]. Figure 7b shows the Raman spectra of the Li-doped NiO/TiO$_2$ calcined catalysts. Differences in band intensity are attributed to the concentration of phases in each support. The anatase and rutile ratios obtained by XRD are reported in Table 2. Furthermore, the bands at 241 (A$_g$), 293 (E$_g$), 350 (E$_g$), 704 (E$_g$), and 770 cm$^{-1}$ correspond to NiTiO$_3$ [44]; these results also agree with the XRD analysis.

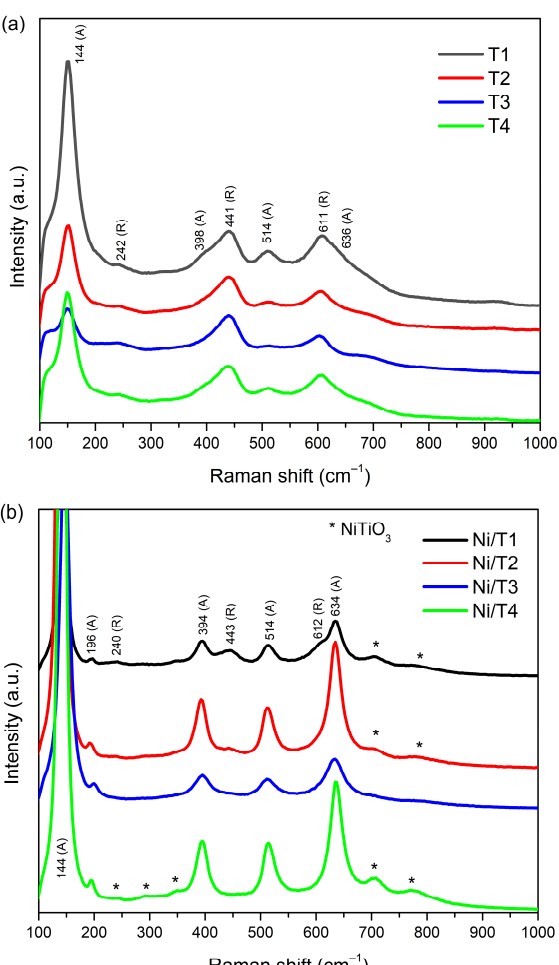

**Figure 7.** Raman spectra of (**a**) $TiO_2$ calcined supports, (**b**) Li-doped $NiO/TiO_2$ calcined catalysts. A = Anatase, and R = Rutile.

### 2.7. Catalytic Activity

Figure 8a–c shows the catalytic activity results during the DRM at 700 °C ($P_{abs}$ = 300 kPa, WHSV = 580 h$^{-1}$). Figure 8a shows that methane conversion was higher (45% initial) in the Ni/T4 catalyst, and the subsequent activity order was similar to that of surface basicity: Ni/T4 > Ni/T2 > Ni/T1 > Ni/T3. Deactivation was noticed in all samples, being more critical with the Ni/T1 and Ni/T3 catalysts, which showed a lower anatase and surface basicity. Figure 8b displays the $CO_2$ conversion profiles; these conversions were higher than $CH_4$ in all cases. Over time, a reduction in $CH_4$ and $CO_2$ activity profiles suggests that carbon deposits block the active sites for $CO_2$ and $CH_4$ activation. This blockage of active sites is not homogeneous; some catalysts are more affected than others (more basic ones are less affected than others). Again, the Ni/T4 catalyst presented a higher activity and stability as a function of the $CO_2$ conversion.

Figure 8c shows the variation in the $H_2/CO$ ratio. As can be seen, the Ni/T2 catalyst gives rise to a higher $H_2/CO$ ratio than all catalysts, at around 0.7. The Ni/T4 sample ratio starts at 0.55 and increases with time to 0.7, and finally decays to 0.6. The remaining samples have an $H_2/CO$ ratio of less than 0.3. Low $H_2/CO$ ratios and high $CO_2$ conversion indicate the contribution of the reverse water gas shift reaction (RWGSR, $CO_2 + H_2 \leftrightarrow CO + H_2O$). According to this reaction, hydrogen is consumed while the concentration of CO increases. Lower $H_2/CO$ ratios ranging from 0.5 to 0.6 have been previously reported for a 15 wt.% $Ni/TiO_2$ system prepared by conventional wet impregnation [45]. Other work has reported an average $CH_4$ and $CO_2$ conversion of ca. 39 and 50%, respectively, and $H_2/CO$ ratios ranging from 0.6 to 0.73, for $TiO_2/Ni$ core–shell catalysts [46]. Thermodynamic analysis

of RWGSR and DRM under reaction conditions confirms the results. The equilibrium conversions of $CH_4$ and $CO_2$ are 63.7 and 76.1%, respectively, and an $H_2/CO$ ratio of 0.82. However, further analysis, including Equations (2)–(5) in the calculations, disagrees with the experimental data and finds that carbon deposits do not form under equilibrium conditions. As discussed above, adding alkali (as lithium in this case) to the catalyst helps prevent carbon formation [8,36,47]. On the other hand, oxygen vacancies promote radical anion $CO_2^{\delta-}$ dissociation into CO and O [48]. The anion $CO_2^{\delta-}$ is formed once the $CO_2$ molecule accepts an extra electron. Therefore, qualitatively, we can infer that the catalysts Li-Ni/$TiO_2$ are a bifunctional system: first, the metal (nickel) provides active sites for $CH_4$ chemisorption and electric transfer to $CO_2$; second, oxygen vacancies are related to strong basic sites where $CO_2$ activation occurs, primarily adsorbed as bidentate carbonates, and finally dissociate on the catalyst surface and also promote the coke gasification due to rapid oxygen migration, helping to increase catalytic activity and stability [49–53].

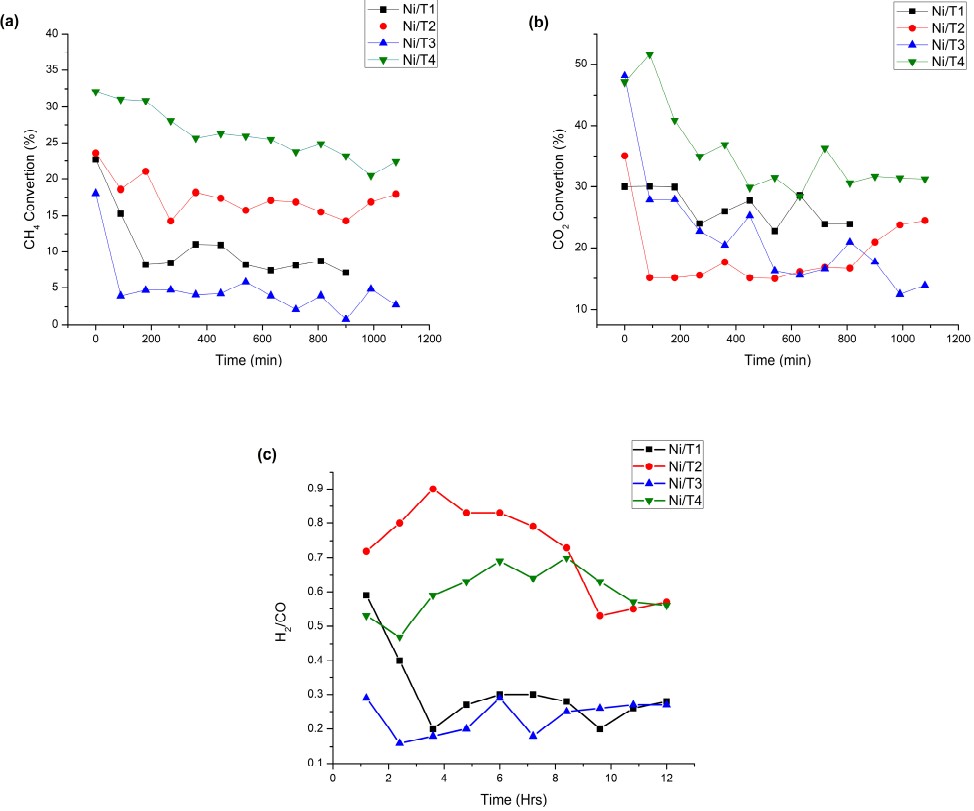

**Figure 8.** Catalytic performance of Li-doped catalyst in DRM (T = 700 °C, Pabs = 300 kPa, WHSV = 580 h$^{-1}$). (**a**) $CH_4$ conversion, (**b**) $CO_2$ conversion, and (**c**) $H_2/CO$ ratio.

The lithium promoter also increases these materials' catalyst stability and basicity. Consequently, the latter facilitates the adsorption of CO and $CO_2$ molecules or inhibits the desorption of $CO_2$ products [35,47,48,54,55]. Previous works have reported Li-doped content ranging between 0.1 and 2 wt.% [47,54], with 0.54 wt.% lithium being the optimal loading in $ZrO_2$-supported Pt catalysts, for low-temperature water–gas shift reactions [54] and 3 mmol/g support (ZnO) being optimal for biodiesel preparation [56], respectively. In this work, Li-loading is ca. 3.4 mmol/g $TiO_2$ support, which is very close to the latter. A synergetic effect exists between basicity, via oxygen vacancies, and doping (Li), so only Ni/T4 and Ni/T2 samples remain with high conversion and stability.

Particle size is another parameter in the DRM reaction; it is not directly tied to basicity. Smaller particles (Ni/T4 < Ni/T2 < Ni/T1 < Ni/T3) present a higher surface area, which is strongly correlated to catalyst activity [57]. The performance of the catalysts in DRM corresponds to the decreasing order of particle sizes. The above results, in CO-DRIFTS

measurements, are not related to the activity of the catalysts; however, this indicates that the support mainly strongly affects the catalytic behavior.

$CH_4$ and $CO_2$ deactivation profiles are multifactorial phenomena that can be attributed to sintering effects, the formation of secondary phases ($NiTiO_3$ and $LiTi_2O_4$), the blockage of active sites due to carbon deposits, the particle size, and the differences in surface basicity between the catalysts. It is the leading property that dictates the chemical behavior of the system. This basicity is governed by these catalysts' anatase content (i.e., the A/R ratio).

### 2.8. Spent Catalyst Characterization

The mass of carbon deposits over the spent catalysts was determined gravimetrically, and the results are reported in Table 6. The following decreasing order of carbon deposition was observed on the catalysts: Ni/T4 < Ni/T2 < Ni/T1 < Ni/T3, which also agrees with DRM performance and surface basicity.

**Table 6.** Quantification of carbon formation obtained gravimetrically and XRD analysis of spent catalysts after DRM (*, not detected).

| Sample | Mass of C [mg] | XRD Analysis (wt.%) | | | | Crystallite Size [nm] |
|---|---|---|---|---|---|---|
| | | Rutile | $LiTi_2O_4$ | Ni | C | Ni |
| Ni/T1 | 2.08 | 67.08 | 21.34 | 7.93 | 3.65 | 41.0 |
| Ni/T2 | 1.4 | 69.78 | 14.51 | 12.02 | 3.69 | 38.2 |
| Ni/T3 | 2.24 | * | * | * | 99.37 | * |
| Ni/T4 | 1.25 | 49.25 | 42.89 | 7.86 | * | 41.0 |

Figure 9 and Table 6 show the diffraction patterns of the spent catalyst after the DRM reaction. Anatase was converted into rutile (PDF 021-1276), the main material in all samples. The presence of lithium is observed in the binary metal oxide $LiTi_2O_4$ (PDF 026-1199, s.g. 227) in the samples Ni/T1, Ni/T2, and Ni/T4. On the other hand, the binary metal oxide $NiTiO_3$, presented on calcined catalysts (Figure 1 and Table 2), was reduced to nickel metal (PDF 004-0850, s.g. 225) and $TiO_2$ as a rutile phase. The metallic nickel in spent catalysts results from in situ reaction conditions (a high temperature and a strong reducing atmosphere); this phase was previously not evident in the CO-DRIFTS chemisorption measurements. Comparing the NiO and Ni crystallite sizes (Tables 2 and 6), it is observed that the latter is higher, evidencing sintering effects, which could further contribute to the catalysts' deactivation. Such metallic nickel exposure overcomes the SMSI effect. Graphitic carbon (PDF 002-0456, s.g. 186) was identified in three samples. The first was presented in Ni/T1 and Ni/T2 catalysts, and the second in Ni/T3 (in this case, almost 100 wt.% is carbon). The Ni/T4 sample shows no crystalline carbon (graphite).

Figure 10 shows the Raman spectra of Li-doped $NiO/TiO_2$ catalysts after the reaction (DRM). The Ni/T1 sample presents bands corresponding to the $TiO_2$ rutile phase at 242, 442, and 608 cm$^{-1}$. The absence of bands related to the $TiO_2$ anatase phase indicates its complete transformation to the most stable rutile phase. The lack of the 2691 cm$^{-1}$ band means the presence of amorphous carbon. Samples Ni/T2, Ni/T3, and Ni/T4, in addition to the bands corresponding to rutile, also show bands at 1335 (D band, disorder in sp$^2$-carbon materials), 1582 (G band, E2g mode), 1606 (D' band, disorder in sp$^2$-carbon materials), 2691 (2D band, graphitic sp$^2$ materials), and 2924 cm$^{-1}$ (D + G band); these are characteristic of carbon materials, specifically graphene [58]. The presence of the 2D band combined with the G-band is indicated by multi-layer graphene [59]. Also, the samples Ni/T1, Ni/T2, and Ni/T4 showed bands at 133 and 664 cm$^{-1}$, corresponding to the $Li_{1+x}Ti_{2-x}O_4$ phase [60]. The multi-layer graphene formation could be explained by the 2D/G bands ($I_{2D}/I_G$) ratio and from the peak position and shape [61]. If the $I_{2D}/I_G$ ratio is ~2 to 3, it is monolayer; $2 > I_{2D}/I_G > 1$ is bilayer graphene and $I_{2D}/I_G < 1$ is multilayer graphene. The $I_{2D}/I_G$ ratio in the Ni/T2, Ni/T3, and Ni/T4 samples are 0.83, 0.53, and 0.56, respectively. Therefore, we can conclude that the graphene produced during the DRM reaction is of the multilayer type.

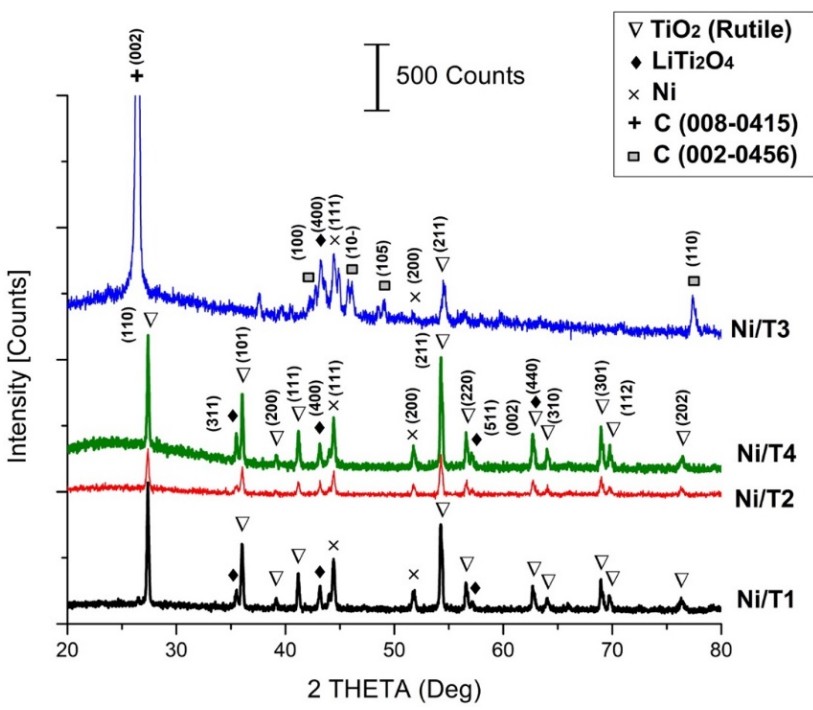

**Figure 9.** XRD patterns of spent catalysts.

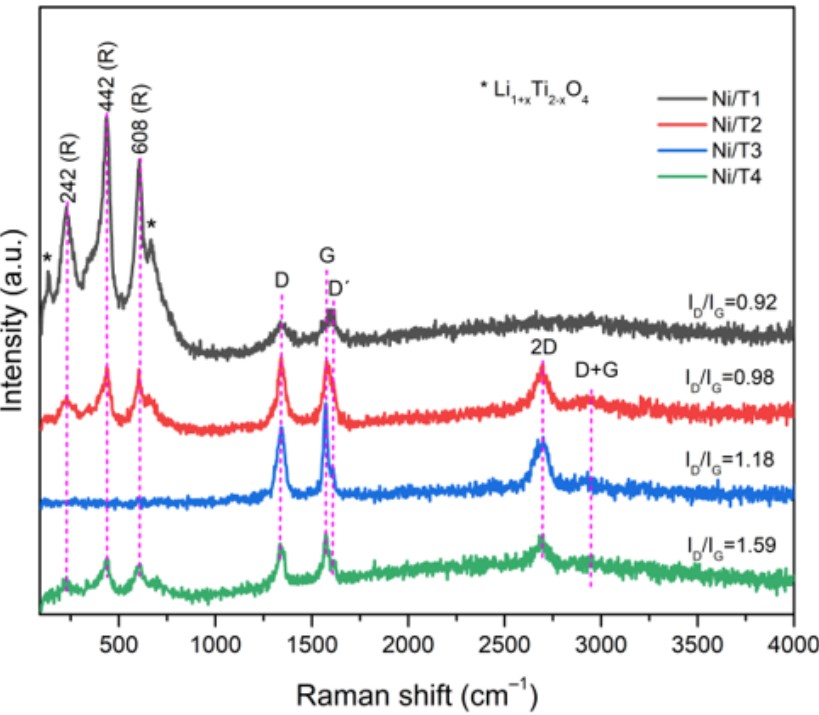

**Figure 10.** Raman spectra of Li-doped NiO-TiO$_2$ catalysts after the reaction.

The intensity of the D band is dependent on the defects, and the G band is present where highly crystalline graphite carbon prevails. Hence, the peak intensities of the Raman spectra were normalized to determine the I$_D$/I$_G$ ratio; this quotient was used to determine the disorder level in carbon atoms and the defect density. Figure 10 shows the intensity ratio of used catalysts: 0.92, 0.98, 1.18, and 1.59 for Ni/T1, Ni/T2, Ni/T3, and Ni/T4, respectively. Figure 11 shows scanning electron micrographs for samples with the lowest and highest activity, namely Ni/T3 and Ni/T4, respectively. As commented before, the Ni/T3 sample evidenced a high density of formed graphitic carbon, which presented as

CNTs, with diameters ranging from 30 to 60 nm. In addition to sintering effects, the CNTs contribute to catalyst erosion until deactivation occurs. On the contrary, the Ni/T4 sample does not show evidence of massive formation of CNTs, but rather of an amorphous (sponge-like) carbon, consistent with XRD and Raman analysis. The amorphous carbon does not contribute to active phase erosion and keeps the catalyst's activity in DRM conditions.

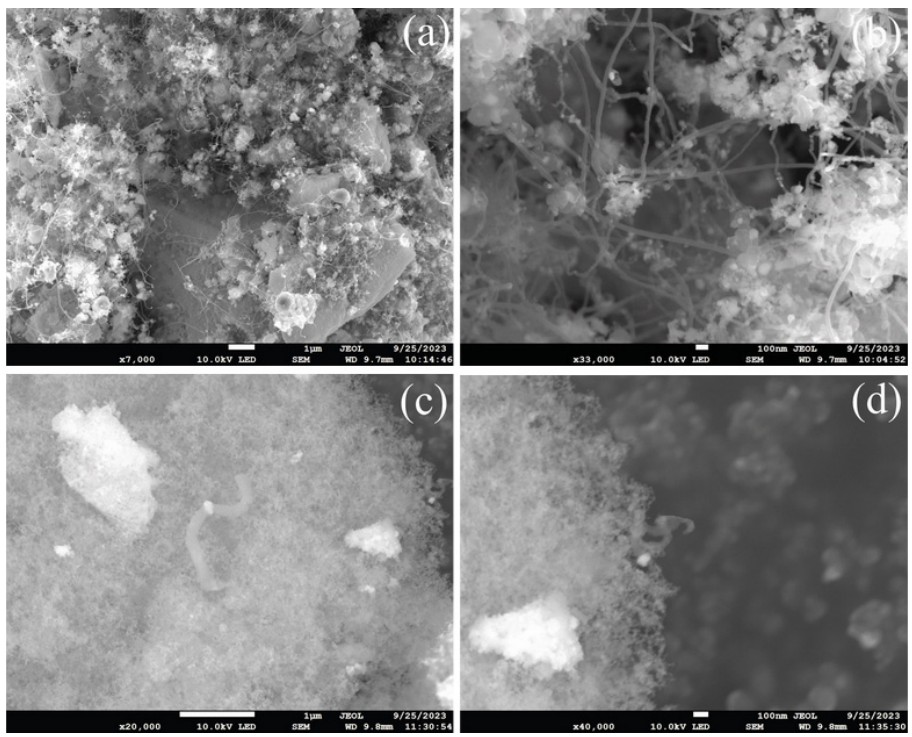

**Figure 11.** FESEM micrographs of catalysts after the reaction: (**a**,**b**) Ni/T3 sample; (**c**,**d**) Ni/T4 sample.

## 3. Materials & Methods

### 3.1. Support and Catalyst Preparation

Mesoporous $TiO_2$ supports were prepared using the sol–gel technique, as reported previously [62]. Firstly, 10 mL of ethanol (Fermont-PQM, MTY, Mexico, ACS reagent) and 2.7 mL of deionized water were dropped in continuous agitation into a titanium (IV) butoxide (Sigma-Aldrich, St. Louis, MO, USA, ≥97 wt.%) $7.56 \times 10^{-2}$ M ethanol-based solution with 1 mL of concentrated HCl (Fermont, ACS reagent). Once the first solution was wholly dropped, stirring continued, depending on the aging time. Several aging times were explored: 2, 3, 4, and 5 days. After, water excess was eliminated by evaporation at 100 °C. After drying, calcination was carried out at 500 °C for two hours. Samples were named T1, T2, T3, and T4, respectively.

Li-doped Ni-based catalysts were prepared using the incipient wetness method. Mother solutions of 0.01 M of nickel nitrate hexahydrate (Fermont, 99 wt.%) and 1.44 M of lithium nitrate (Fermont, 99 wt.%) were used, with as a precursor of metallic nickel as the active phase and lithium as a support promoter, respectively. A defined amount of nickel and lithium precursor solution was utilized to obtain 15 and 2 wt.%, respectively. The solution was heated to 125 °C in the nickel case and added to the $TiO_2$ solid support. Then, the water excess was eliminated by evaporation at 75 °C overnight. After drying, the impregnated support was calcinated at 500 °C for four hours to form nickel oxide ($Ni/TiO_2$). Samples were named Ni/T1, Ni/T2, Ni/T3, and Ni/T4, respectively. Once the nickel oxide phases were obtained, a similar procedure was implemented for drying and calcination using the Li mother solution.

### 3.2. Characterization

The structural characterization of all catalysts was performed by powder XRD. Observed patterns were obtained using a Rigaku Ultima IV diffractometer, operated at 40 kV and 44 mA in Bragg-Brentano geometry, with a Cu-target X-ray generator ($Cu_{K\alpha}$ = 0.15419 nm). Diffraction patterns were collected from 20° to 80° in 2θ, with a step size of 0.02° and a scan rate of 0.2° min$^{-1}$.

Quantitative phase analysis of fresh and spent catalysts was realized using the Full-Prof software (https://www.ill.eu/sites/fullprof/, accessed on 19 October 2023) [63]. A crystallographic model was obtained from phase identification. Hence, CIF files (*.cif) for fresh catalysts were 647,553, 9852, 33,854, and 9866 from rutile, anatase, nickel (II) titanate, and bunsenite phases. On the other hand, Rietveld refinements of spent catalysts used the following CIF files: 647,553, 15,789, 52,331, and 31,170 for rutile, lithium titanate, nickel, and carbon, respectively.

Crystallite sizes were obtained by the Scherrer equation [64], considering an instrumental broadening of 0.1308° between 25 and 50° in 2θ, obtained with a NIST Si-640c standard reference material. The morphology of fresh and spent catalysts was studied with a field emission scanning electron microscope, JEOL JSM 7100F, operated at 10 keV, coupled with a lower electron detector (LED). The $H_2$ temperature-programmed reduction ($H_2$-TPR) of catalysts was performed on a Bel Japan Belcat-B. Samples of fresh catalysts were weighed (50 mg) and treated at 250 °C for three hours. The reducing gas contained 5% $H_2$ (balanced with Ar) at a 50 mL min$^{-1}$ flow rate and a heating rate, $\beta$, equal to 10 °C min$^{-1}$. Before the TPR analysis, the $H_2$ consumption was calibrated using a known amount of NiO. Only one peak (TCD signal) was obtained and integrated to get the $H_2$ consumption (μmol/g) to area ratio, which was used for further quantitative TPR analysis.

Basicity of materials (basicity–DRIFTS) was determined by MBOH conversion into acetone and acetylene using a spectrophotometer, Thermo Scientific Nicolet IS-50, operated in absorbance mode, with DRIFTS in the range between 400 and 500 cm$^{-1}$, using a stainless steel dome with KBr windows and a Harrick Praying Mantis. Firstly, catalyst activation was performed in situ: a mixture of catalyst (10 wt.%) and KBr was finely prepared and then placed into a Praying Mantis accessory. A slow heating rate of 5° min$^{-1}$ was used up to 550 °C, with a constant $H_2$ flow of 40 mL min$^{-1}$. The reduction temperature was maintained for one hour. Secondly, 100 μL of MBOH solution was added to each sample to saturate the basic sites in the catalyst. Measurements were conducted with 32 scans per sample. After each measure, an Ar flow was used to clean the catalyst's surface.

CO–DRIFTS chemisorption analysis was performed using a similar experimental setup as the basicity–DRIFTS measurements. Before chemisorption analysis, catalysts were reduced in $H_2$, at 550 °C. After the reduction, an inert flow of $N_2$ was injected until cooled to room temperature. For the chemisorption test, 100 mL min$^{-1}$ of CO (10 *v/v* %, balanced with Ar) was injected into the Praying Mantis dome, which was heated to 200 °C. The spectrometer was operated at a resolution of 0.09 cm$^{-1}$ and performed 64 scans per sample.

The particle size of materials was measured using dynamic light scattering characterization (DLS), using a Micromeritics Particle Analyzer Nanoplus 3, equipped with a 660 nm diode laser with a maximum power of 100 mW and an avalanche photodiode detector. A disposable plastic cuvette of 0.9 mL was used in each measurement. Measurements of surface area and textural analysis of fresh catalyst powders were carried out using Quantachrome NovaTouch. Before the study, the samples were degassed at 150 °C for five hours under a $N_2$ flow.

Raman spectra were collected using a Renishaw InVia microscope, using a 532 nm green laser and coupled with a holographic filter. Spectra were measured using 10% of the laser power, five accumulations, and 10 s.

### 3.3. Activity Test in Dry Reforming of Methane

The activity and stability of the catalysts were determined in the DRM. Reactions were carried out at 700 °C (heating rate = 10 °C min$^{-1}$) using 50 mg of fresh catalysts

placed on quartz wool inside a tubular reactor (ID = 7.74 mm, L = 500 mm). Activation was performed in situ at 550 °C, using a constant gas flow of 35- and 80-mL min$^{-1}$ of $H_2$ and Ar, respectively. In DRM, the reactants ($CO_2$ and $CH_4$) were fed at an equimolar ratio with a 30 mL min$^{-1}$ flow, diluted with Ar at 80 mL min$^{-1}$. The reactants were fed into the reactor at 200 kPa (manometric pressure). The temperature was monitored with a thermocouple situated outside the catalyst bed. The overall reaction time was 12 h. Activity evolution was monitored using a gas chromatograph, namely Fuli Instruments 979011, equipped with a TCD detector and a packed silica gel column, using 25 mL min$^{-1}$ of $N_2$ as carrier gas. The temperature in the oven was 100 °C, and the detector was set at 120 °C and 40 mV.

## 4. Conclusions

In this study, we successfully synthesized mesoporous $TiO_2$ supports using the sol–gel method with varying aging times, followed by the deposition of Ni. Our microstructural analysis revealed the coexistence of rutile and anatase supports within the catalysts. To assess the concentration of Lewis basic sites on the catalyst surface, we relied on the reliable FTIR technique, tracking the conversion of MBOH into acetone and acetylene. Experimental results indicated that the basicity of Li-doped Ni/$TiO_2$ catalysts was predominantly governed by the anatase content and the A/R ratio, closely related to oxygen vacancies. The order of basicity observed in our study was as follows: Ni/T4 > Ni/T2 > Ni/T1 > Ni/T3; the longest aging time corresponds to the highest degree of basic character across the support surface.

Notably, a synergistic interplay between basicity and lithium doping was discernible, which prominently manifested in the exceptional conversion rates and stability exhibited by the Ni/T4 and Ni/T2 samples. Furthermore, we found a strong correlation between basicity and the conversion of $CH_4$ and $CO_2$ in the dry reforming of methane (DRM). Field emission scanning electron microscopy and Raman analysis provided compelling evidence for the formation of graphitic multilayered carbon, a phenomenon inversely proportional to basicity and directly associated with catalyst stability. These insights advance our understanding of the crucial role of basicity in catalytic performance and provide valuable guidance for catalyst design and optimization.

**Author Contributions:** Conceptualization, F.P.-A. and M.A.V.; Methodology, E.R.-G., M.A.V. and F.P.-A.; Validation, M.A.V.; Formal analysis, E.R.-V.; Investigation, E.R.-V. and E.R.-G.; Writing—original draft, V.P.-M. and F.P.-A.; Writing—review & editing, V.P.-M., E.R.-V., E.R.-G., M.A.V. and F.P.-A.; Project administration, F.P.-A.; Funding acquisition, F.P.-A., and E.R.-V. All authors have read and agreed to the published version of the manuscript.

**Funding:** This work was supported entirely by the project SEP-CONACYT 2016-286940.

**Institutional Review Board Statement:** Not applicable.

**Informed Consent Statement:** Not applicable.

**Data Availability Statement:** No new data were created or analyzed in this study. Data sharing is not applicable to this article.

**Acknowledgments:** The authors thank the *Universidad Politécnica de Chiapas* for FESEM analysis via the CONAHCYT-322405 project.

**Conflicts of Interest:** The authors declare no conflict of interest.

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
