# Peer review of "Dry Reforming of Methane over Li-Doped Ni/TiO2 Catalysts: Effect of Support Basicity"

_methane, doi:10.3390/methane2040031_

Round 1

Reviewer 1 Report (Previous Reviewer 1)

Comments and Suggestions for Authors

The article entitled “Dry reforming of methane over Li-doped Ni/TiO2 catalysts: Effect of support basicity” shows the use of Li as a dopant of a TiO2 support prepared by the sol-gel method varying the aging time. The author said that long aging times favor the basic character caused by the oxygen defects in TiO2, which is intrinsically related to catalyst performance, stability, and carbon formation during the reaction. The manuscript remarks a deactivation process that reduces the catalytic performance of the catalysts. Despite the activity is not so high the conclusion of the article is interesting and the effect of Li nanoparticles on catalytic surface is showed. Although this work is quite interesting there are issues the author should address for the draft to be accepted.

·         The TPR conditions and description are missing in experimental part.

·     For TPR the total H2 consumption should be included. The reduction ratio should be calculated to clarify the effect of the metallic particles size on the reducibility properties and on activity.

·    In XRD results, a solid solution between Li and NiO phase should be taken accounts in the discussion and their influence on activity results should be clarified. The anatase/rutile effect should be related with the formation of solid solutions and therefore, oxygen vacancies.

·  The relation between particle size and activity is not clear. What happened with size and its effect in activity on Ni/T3? TEM images could help to clarify the real size of the Ni nanoparticles in all solids.

·    The author said: A synergetic effect exists between basicity via oxygen vacancies and doping (Li), so only Ni/T4 and Ni/T2 samples remain with high conversion and stability.  However, the discussion about the acidity/basicity properties and the surface properties on activity should be improved since the influence on textural properties it seems to be less important, and the crystalline sizes are similar but the difference in basicity properties are important.  

·   Moreover, images of carbon deposits on spent catalysts should be included to demonstrate the type of carbon formed and its influence on deactivation process.

Comments on the Quality of English Language

Minor editing of English language required

Author Response

Reviewer 2 Report (New Reviewer)

Comments and Suggestions for Authors

This paper titled" Dry reforming of methane over Li-doped Ni/TiO2 catalysts: Effect of support basicity" needs a serious major revision :

1- In the abstract section, the first 3 lines must be removed as the given information is general and can be moved to the introduction section

2- In the abstract section, must contain some numeric results from the obtained ones in this work.

3- All given images are poor in quality and the resolution must be increased to be more clear for the readers and included writings must be clear also.

4- The XRD patterns must be indexed showing the Miller indices and (h,k, l) values, and refinement analysis with all parameters must be also given ( space groups, cell parameters, refinement parameters, .......etc.,)

5- The conclusion must include the potential of this work and the limitations as well.

Comments on the Quality of English Language

It needs minor corrections and grammatical mistakes must be checked 

Round 2

Reviewer 1 Report (Previous Reviewer 1)

Comments and Suggestions for Authors

The manuscript can be accepted in current form

Comments on the Quality of English Language

Minor editing of English language required

Reviewer 2 Report (New Reviewer)

Comments and Suggestions for Authors

The authors have done all corrections and editing as well and the paper is now can be accepted for publications.

This manuscript is a resubmission of an earlier submission. The following is a list of the peer review reports and author responses from that submission.

Round 1

Reviewer 1 Report

Comments and Suggestions for Authors

     This work correlates the effect of TiO2 (support) aging time on surface properties and the corresponding catalytic activity and stability in the DRM, keeping constant the composition of the active phase (Ni particles) and promoter (Li). Although this work is quite interesting there are issues the author should address for the draft to be accepted.

·         The authors affirm that the basic character is directly influenced by the anatase content of Li-doped Ni/TiO2 catalysts. The longest aging time agrees with the highest basic character associated with oxygen defects over the support surface. In this part the authors describe the results but there is neither further elaboration, nor comparison with current literature.

·         In this line the comparison with previous results is mandatory in many parts of the text. Effect of the addition of promoters on activity have been studied several times for the reaction studied. In a same way the comparison and discussion with similar bi- metallic catalysts is necessary.

·         It is also mandatory to clarify the role of Li on the reaction mechanism and on physicochemical properties. Do these metals form an alloy, solid solution or the activity depends on the cooperative effect or oxygen spillover effect improvement? This should be better clarified.

·         In the same way the influence of the preparation method on activity and selectivity is not well explain. The author should clarify if there are differences if the metals are added in a different way on the properties of the solid.

·         Additionally, the pore diameters and the isotherms of the adsorption / desorption should be included in the results. These results should be related with the dispersion data, particle sizes, and metallic crystallite sizes.  How these results affect the catalytic response of the catalytic system?

·         H2-TPR analysis is mandatory simultaneously the H2 consumption and reducibility percentage of the solids should be also included. The reduction ratio should be calculated to clarify the effect of the metallic particles size on the reducibility properties and on activity. The author should include these results. In a similar way the XRD results for the catalysts after calcination should be better discussed.

·         The ratio anatase/rutile phases should be included. It is mandatory to clarify the effect of these phases on activity properties.

·         TEM images are necessary. Additionally, the authors should include a particle size distribution figure to be confident with the result discussed about the small size of the metallic nanoparticles and the influence of Li addition on these characteristics.

·         Can the author see the same metallic size particles in the catalysts after reaction? A full characterization might help the author to increase the discussion.

·         Why a so high Ni loading? The author must clearly state their choice and additionally the justification of the amount of Li used is required.

·         According to the activity results there is a decay on the activity. The authors must explain this behavior and the relation with the RWGS and /or deactivation of the catalysts. It is also necessary to show the activity in the same period of time. This discussion is important to demonstrate the activity of the solids studied and to be confident with the conclusion.

·         For this article the deactivation under operational conditions is an important issue. However, the results are confused since the deactivation problem is not well justified. They should justify and explain better the novelty of these results, the causes of deactivation and how it can be decreased. The authors should answer: What kind of deactivation is expected? Which kinds of intermediates are formed?

·         They should justify and explain better the novelty of these results: selectively etc, the causes of deactivation and how it can be decreased, advantages (disadvantages) of bimetallic catalysts.

·         In short, even though the premise of this paper is very interesting, the discussion and justification of the parameters and results must really be improved to reach enough quality to be accepted.

Reviewer 2 Report

Comments and Suggestions for Authors

The manuscript "Dry reforming of methane over Li-doped Ni/TiO2 catalysts: Effect of TiO2 aging time" by V. Pérez-Madrigal et al. reports a study focused on dry reforming of methane. And the effects of catalyst preparation conditions on the physicochemical properties and catalytic performance of Li-doped Ni/TiO2 are also discussed in detail. The subject is in the scope of MDPI. I recommend it for publication after a major revision. The comments are listed as follows:

1. The authors emphasize that the base Lewis sites play an important role in dry reforming of methane. However, in this manuscript, the authors only give the concentration of the base Lewis sites. The strength of surface base also needs to be detected by CO2-TPD.

2. The authors mainly provide the characterization results of the catalyst support TiO2 in detail. However, Ni as the active center for methane activation also should be analyzed by H2-TPR and TEM.

3. Due to the high reaction temperature, loss of active components, such as Li, may occur. The authors should give the catalyst composition before and after reaction.

4. Some mistakes must be corrected, such as subscript (cm-1), and figure 6.

Reviewer 3 Report

Comments and Suggestions for Authors

The paper is mostly a report on experimental results. Indeed it seems that Li and aging of TiO2 modify the catalytic performance and the basicity measured by MBOH is on line with this. But as obtained the results are overinterpreted, particularly lines 284-285 "The longest aging time agrees with the highest basic character associated with oxygen defects over the support surface." How these "oxygen defects are measured in the present paper?

Some assertions like "the chemical behavior of TiO2 leads to Lewis base character due to the presence of O-2 sites on its surface" are not well written and one wonders if the concept has been understood: This is the Lewis base character which determines its chemical behavior, and not the reverse. O2- sites on surface (and in bulk) are present in any oxides. Are the  slightly basic O2- sites or oxygen vacancies important?

Line 228 "Basicity is governed by the anatase content in these catalysts".  The structures and cation sites are very similar, why would it be? What is the basicity of rutile (measured by MBOH)?

Line 227-228 "Particle size is another parameter in DRM reaction; it is linked not directly to basicity but to surface energy". What is the meaning of surface energy 

Line 230 "Catalysts’ performance in DRM agrees in increasing order in particle size". But: the first important property to measure is the specific surface area, which probably shrinks during the reaction, and also the porosity (if any), not only the size of particles

What is the role of NiTiO3 or LiTi2O4 on the catalyst performance?

Round 2

Reviewer 1 Report

Comments and Suggestions for Authors

The revised version shows confused concepts and not well justified. I do not recomened it for publication.

Reviewer 2 Report

Comments and Suggestions for Authors

The revised manuscript can be considered to be accepted for publication in this journal.